# Modeling Shallow Landslide Susceptibility and Assessment of the Relative Importance of Predisposing Factors, through a GIS-Based Statistical Analysis

**Massimo Conforti** [1,*] and **Fabio Ietto** [2]

1 National Research Council of Italy—Research Institute for Geo-Hydrological Protection (CNR-IRPI), Via Cavour 4/6, 87036 Rende, Italy

2 Department of Biology, Ecology and Earth Science, University of Calabria, Via P. Bucci, cubo 15/B, 87036 Rende, Italy; fabio.ietto@unical.it

* Correspondence: massimo.conforti@irpi.cnr.it

**Abstract:** Shallow landslides are destructive hazards and play an important role in landscape processes. The purpose of this paper is to evaluate the shallow landslide susceptibility and to investigate which predisposing factors control the spatial distribution of the collected instability phenomena. The GIS-based logistic regression model and jackknife test were respectively employed to achieve the scopes. The studied area falls in the Mesima basin, located in the southern Calabria (Italy). The research was based mainly on geomorphological study using both interpretation of Google Earth images and field surveys. Thus, 1511 shallow landslides were mapped and 18 predisposing factors (lithology, distance to faults, fault density, land use, soil texture, soil bulk density, soil erodibility, distance to streams, drainage density, elevation, slope gradient, slope aspect, local relief, plan curvature, profile curvature, TPI, TWI, and SPI) were recognized as influencing the shallow landslide susceptibility. The 70% of the collected shallow landslides were randomly divided into a training data set to build susceptibility model and the remaining 30% were used to validate the newly built model. The logistic regression model calculated the landslide probability of each pixel in the study area and produced the susceptibility map. Four classification methods were tested and compared between them, so the most reliable classification system was employed to the shallow landslide susceptibility map construction. In the susceptibility map, five classes were recognized as following: very low, low, moderate, high, and very high susceptibility. About 26.1% of the study area falls in high and very high susceptible classes and most of the landslides mapped (82.4%) occur in these classes. The accuracy of the predictive model was evaluated by using the ROC (receiver operating characteristics) curve approach, which showed an area under the curve (AUC) of 0.93, proving the excellent forecasting ability of the susceptibility model. The predisposing factors importance evaluation, using the jackknife test, revealed that slope gradient, TWI, soil texture and lithology were the most important factors; whereas, SPI, fault density and profile curvature have a least importance. According to these results, we conclude that the shallow landslide susceptibility map can be use as valuable tool both for land-use planning and for management and mitigation of the shallow landslide risk in the study area.

**Keywords:** shallow landslide; landslide predisposing factors; logistic regression model; susceptibility map; calabria

## 1. Introduction

Shallow landslides are phenomena triggered by heavy rainstorms with short duration and intense precipitation or rainfall with long duration and medium–low intensity [1–3]. These phenomena, in mountain and hillslope environments, represent one of the most important denudation processes and can affect human life and infrastructure both directly and indirectly [4–7]. The spatial distribution of shallow landslides is strongly influenced

both by different climatic conditions and by environmental framework including weathered rock grade, soil characteristics, land use/cover, and morphological features [3,8–10]. Furthermore, in recent decades, the shallow landslide occurrence seems to be increasing due to climate changes [11,12] and several countries, including Italy, were identified as high landslide risk areas [6,13–15].

Italian territory is largely affected by shallow landslide events mainly because of its climatic and geological features. Anyway, in Italy instability phenomena are often linked to uncorrected land use management as well. The Calabria region in Southern Italy, is periodically affected by shallow landslide events, which cause considerable damage to buildings and infrastructures posing serious threat to the population and economic loss [7,16–23].

The construction of landslide susceptibility maps becomes a useful tool both to better identify areas prone to shallow landslides and the interaction between the last one areas and the geo-environmental features able to control the instability processes. Consequently, landslide susceptibility zoning represents the first step for the evaluation of landslide risk and also supply important information for decision and land planning [24,25].

The reliability of landslide susceptibility maps mainly depends on the quantity and quality of available data (e.g., landslide inventory and landslide predisposing factors), the working scale and the use of the appropriate analysis methodology [25–27]. In recent decades, with the increasing use of the geographic information system (GIS) techniques, various methods were employed for landslide susceptibility mapping, such as: heuristic, deterministic, statistic, and machine learning approaches [28–32]. Heuristic approaches, for instance analytic hierarchy process (AHP) or the weighted linear combination (WLC), are commonly used to evaluate landslide susceptibility [25,29,30,33–35]. These approaches are subjective methods based on the prior knowledge of the terrain condition. Thus, the expert geomorphologists identify landslides, ascertain the main influencing factors and recognize sites that have similar environmental characteristics [25,30,36,37]. Deterministic methods employ physical laws and engineering principles of stability analysis to estimate safety [28,38]. The deterministic methods are generally applied when the study area is small and the geologic properties are relatively homogeneous in the investigated area [39,40]. Statistically-based landslide susceptibility models are based on the relationships between dependent (landslides) and independent variables (predisposing factors) [41]. In particular, the statistical models examine the combination of parameters that have led to landslides in the past [24,26,31]. Hence, quantitative predictions of the susceptibility are made by identifying areas where similar background conditions occur. Several statistical approaches, such as: frequency ratio [30,42–45], weights of evidence [22,43,45,46], certainty factors [46,47], evidential belief function [48,49], statistical index [43,50,51], index of entropy [52,53], logistic regression [18,29,42,45,51,54], and multivariate adaptive regression spline [54,55] were frequently used for landslide susceptibility mapping, producing reliable results. In addition, in the last decade, machine learning algorithms were frequently applied for landslide susceptibility analysis, such as: artificial neural network [29,32,42,45,56,57], maximum entropy [54,58,59], random forest [44,60,61], support vector machine [32,57,59,62–64], naïve Bayes [41,62,65], neuro-fuzzy [63,66], boosted regression tree [44,67], kernel logistic regression [57,68], decision trees [45,63], classification and regression tree [44,54,61], and alternating decision trees [68].

The spatial distribution of landslides is linked to the interaction of various geo-environmental factors, such as geological, soil, land cover/use, morphological, hydrological and climatic features, which can contribute in different manner for the landslide susceptibility modeling in a territory [69–72]. A review of the literature shows that there is no consensus on the choice of predisposing factors for landslide susceptibility modeling [30,59,73–75]. In several study the predisposing factors have selected on the basis both of the characteristics of the investigated area and of data availability [22,30,41,56,76]. More recently, several authors—e.g., [70,77]—tried to select the more appropriate and effective predisposing factors among the geo-environmental ones available, because not all

predisposing factors have an equal importance to evaluate the landside prone-areas. Thus, the most important predisposing factors must be identify, excluding from the analysis the least effective because they can reduce the predictive capability of the models.

In this context, the main goals of this research can be summarized as following: (i) to produce a shallow landslide susceptibility map, employing the GIS-based logistic regression model; (ii) to understand the predisposing factors that mainly control the spatial distribution of the shallow landslides.

Logistic regression is a statistical method widely used to parameterize a non-linear relationship between dependent and independent variables [29,42,45,54]. It is largely employed if the dependent variable has a binary or dichotomous output. The method also allowed to assess the relationship between the dependent and independent variables, which can be continuous and categorical [29,45,54]. Furthermore, the independent variables have not necessarily a normal distribution [45,47]. Thus, the features of the logistic regression model make it as an ideal choice for modeling shallow landslide susceptibility.

The study area falls within the Mesima basin, located in the southern Calabria (Italy), characterized by geological, morphological, and climatic features that are common in many areas on Calabria region.

## 2. Study Area

The Mesima basin extends for a surface of 806.4 km² in Calabria region (southern Italy) and spanning from 38°42′35″ N to 38°19′35″ N of latitude and from 15°55′17″ E to 16°19′35″ E of longitude. (Figure 1). The Mesima basin represents the main fluvial catchment of the central part of the Calabria and its mouth flows in the Tyrrhenian sea on the south area of the Capo Vaticano promontory (Figure 1). The basin area is dominated by a hilly landscape on the central and western side, while, high mountains of Serre and Aspromonte Massifs characterize the eastern border. Elevation in the area ranges from 0 m to 1275 m a.s.l., with an average value of 400.5 m (standard deviation = 287.7 m). Slope gradient, computed from a 5 × 5 m DTM, ranges from 0° to 78.2°, with a mean value of 13.8° and a standard deviation of 11.1° [78].

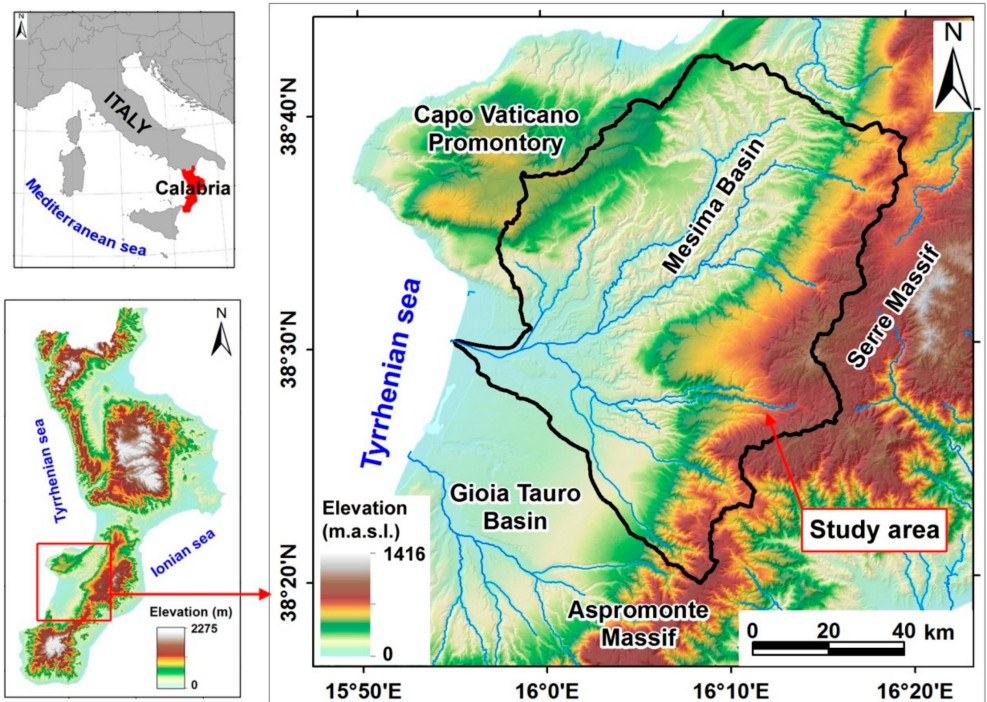

**Figure 1.** Location of study area.

The annual precipitation ranges between 780 and 1890 mm/year, proceeding from the coast toward the internal mountainous area, with an average annual value of about 1140 mm. Rainfalls are most abundant from October to March, during which landslides and water erosion processes are common [7,21,78,79].

From geo-structural point of view, the study area is located within the central sector of the Calabrian–Peloritan arc (CPA) [80,81]. The CPA is a portion of the central Mediterranean orogen extruded from the oceanic crust of the Ionian Basin during the final phase of the Africa–Europe collision [82]. The CPA is constituted by nappes of Jurassic to Early Cretaceous ophiolite-bearing sequences, overlying Hercynian and pre-Hercynian crystalline basement, partially affected by Alpine metamorphism [83]. Cenozoic sedimentary sequences cover part of the oldest terrenes [80,84].

The geodynamic and tectonic evolution of the CPA is mainly related to a prevalent compressional tectonic movement linked to the subduction process involving the African paleomargin [85]. Since the Lower-Middle Pleistocene, extensional processes, represented by NE–SW, N–S, and E–W tectonic lineaments [85–87], were accompanied by a strong regional uplift [88–90]. The high uplift rates are responsible both of reliefs with high erosive energy and of continuous rejuvenation of the hydrographic network [21].

The lithology of the Mesima basin area [80] is dominated by Plio-Pleistocene sedimentary deposits that partially cover the Paleozoic plutonic rocks or directly lie on the Miocenic sedimentary succession. Silty clay deposits of batial environment, overlain by sandwave deposits and sandy turbidite (Pliocene in age), characterize the upper sedimentary sequence that, locally, is covered by a Pleistocenic conglomerate. The lower sequence is constituted by evaporitic limestones (Miocene in age) that overlap, in discontinuity, on the crystalline basement make up of gneiss and granite rock masses (Paleozoic in age). The crystalline basement is involved in ancient severe weathering processes [91,92], causing widespread landslides [10,93]. Finally, Quaternary deposits, constituted by eluvial–colluvial layers (Holocene in age), unconformably overlies on the sedimentary marine deposits or, directly, on the crystalline basement.

The morphology of the study area is mainly controlled by structural setting, lithology, and attitude of bedding [78,94]. Indeed, in the areas constituted by soft sediments the landscape is characterized by rounded hilly morphology and wide valleys; while, in the areas dominated by harder rocks the landforms are more rugged and the valleys are narrow and deep. Along the main valleys of the basin, many orders of fluvial terraces were developed [94].

In according to the USDA classification [95], the soils developed in the studied area were classified mainly as Inceptisols and Entisols and secondly by Alfisols [96,97]. Soil depth range from few decimeters to more than 1.5 m and the pedoclimatic regime is xeric and thermic, shifting to udic and mesic in the mountain area [96].

The morpho-structural setting, the high weathering processes affecting the crystalline rocks and the climate regime make the studied area prone to intense erosion and landslide phenomena, both shallow and deep-seated [7,78,88,98].

## 3. Materials and Methods

The main process for the construction of a landslide susceptibility model is the gathering and designing of a geo-spatial database, which is constituted essentially by two datasets [69]. The first dataset represents the landslide inventory map. The second one is related to landslide predisposing factors. Landslide susceptibility modeling needs a clear understanding of the relationship between the spatial distribution of the landslides and the landslide predisposing factors [24,31]. Data collected and methodologies used in the present study were schematically summarized in Figure 2, which includes the following steps: (1) preparing a shallow landslide inventory map; (2) selecting appropriate landslide predisposing factors; (3) modeling shallow landslide susceptibility; (4) assessing relative contribute of the landslide predisposing factors; (5) generating susceptibility map; (6) assessing prediction accuracy of the susceptibility model and related map.

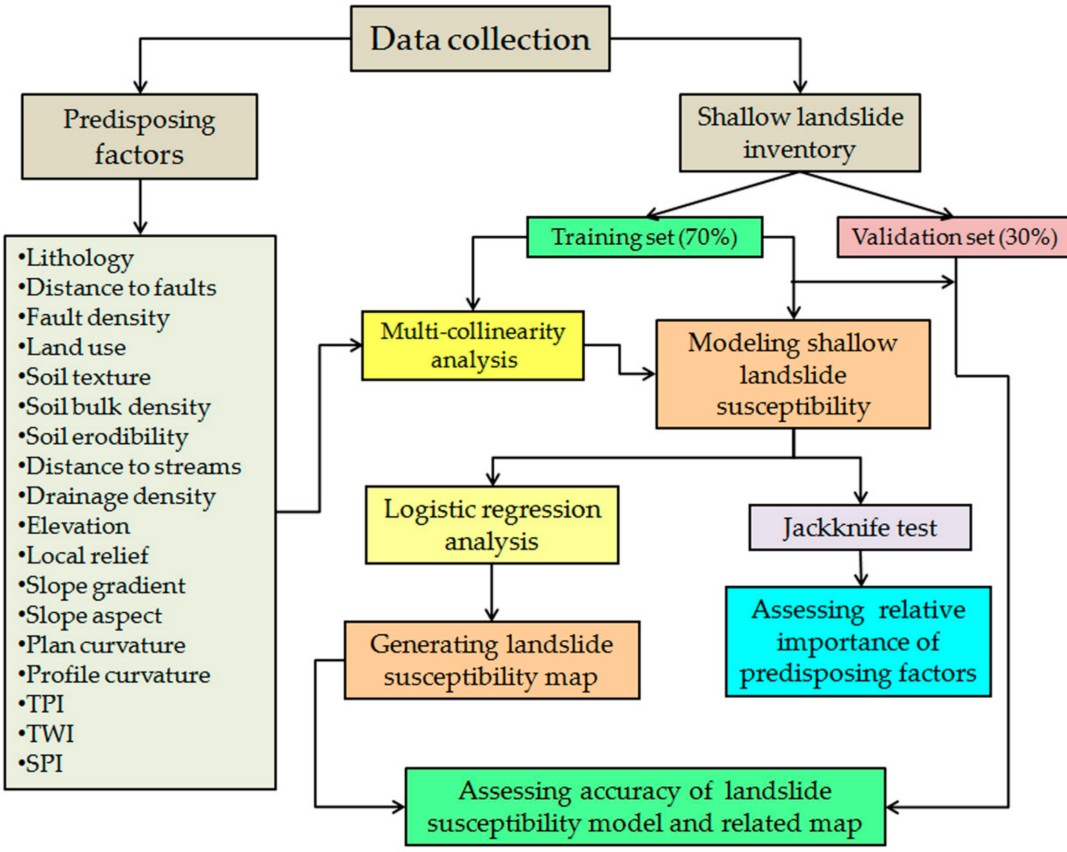

**Figure 2.** Schematic flowchart adopted in the current study.

QGIS 3.16 software was employed for the management and collection of the data, as well as to carry out the shallow landslide susceptibility map; while the logistic regression analysis was performed using the IBM SPSS Statistics software (v. 26).

### 3.1. Shallow Landslide Inventory Map

The quality of the landslide inventory affects the results of landslide susceptibility prediction. This study uses shallow landslides triggered mainly by intense rainfall events in the time ranging between 2000 and 2019. The landslide inventory map (Figure 3a) was prepared through a combination of orthophotos, dating to 2001 (1:10,000 scale) and to 2008 (1:5000 scale), downloaded from the website of the Cartographic Center of Calabria region (http://geoportale.regione.calabria.it/opendata, accessed on 30 November 2020) and Google Earth satellite images available from 2010 to 2019. In addition, several field surveys were carried out between 2019 and 2020 in order to integrate and validate the data obtained by interpretation of the orthophotos and satellite images. The recognized shallow landslides were mapped in a point format, representing the center of the source area. A total of 1511 shallow landslides were identified, corresponding to a landslide density of about 1.9 landslide/km$^2$. The landslide area ranges from few square meters (about 20 m$^2$) to more than 1000 m$^2$; whereas, the landslide width and length range from 5 to 20 m and from 5 to 70 m respectively. In the study area, field observation revealed that landslide types are mainly earth-slide, earth-flow, and slide-earth flow [99]. Slide-earth flows are landslides that trigger as slides and then evolve into flows. Therefore, they show the characteristics of slide types (most commonly rotational) in the upper part, whereas the transport and accumulation areas are more similar to earth flows. In some events, the accumulation zones ending with fan-shaped toes. In addition, it was observed that the failure surface of the shallow landslides was commonly located at the soil-bedrock contact

(2–3 m deep), sometimes involving portions of the weathered bedrock. In many cases these shallow landslides caused damage to many buildings and roads (Figure 3b).

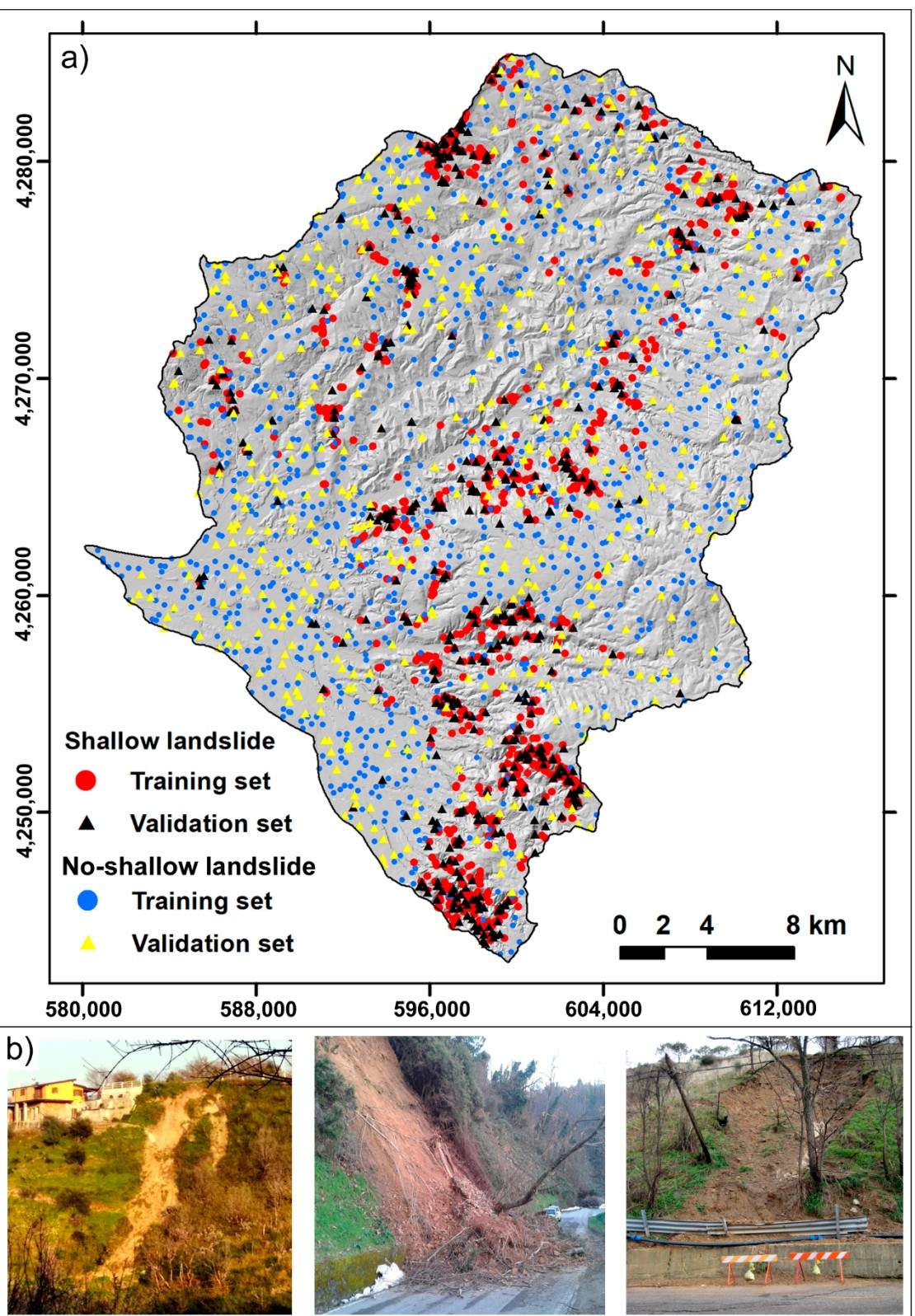

**Figure 3.** (**a**) Map of shallow landslide and no-shallow landslide points; (**b**) Examples of shallow landslides in the study area.

The split of the dataset is of importance to verify the performance of the susceptibility model [100]. Based on existing literature reviews e.g., [29,44,46,56,59,71], to construct and validate the shallow landslide susceptibility model, the landslide inventory was randomly partitioned into two groups: a training set (70%, equal to 1058 landslides) and a validation set (30% equal to 453 landslides). In addition, an equal number (1511) of non-landslide point locations were randomly created in areas without shallow landslides; hence, the 70% and the 30% of the points were allocated to the training and verification datasets, respectively. Values equal to 0 and 1 were assigned to non-landslide and landslide areas respectively. Furthermore, the training dataset was also used to identify the appropriate landslide predisposing factors for the susceptibility analysis.

### 3.2. Shallow Landslide Predisposing Factors

The occurrence of shallow landslide phenomena is closely influenced by the interplay of various geological, topographic, hydrological, and environmental factors [19,44,69,101–103]. Therefore, an important step to assess the landslide susceptibility is the selection of the predisposing factors, the understanding of their combined effects and the relative importance of the selected factors on landslides distribution. There is no common guiding principle for selecting the predisposing factors [30]. On the basis of the literature review, of the characteristics of the study area and of data availability, a total of 18 predisposing factors were considered in the study area for the analysis of shallow landslide susceptibility (Table 1). The predisposing factors include lithology, distance to faults, fault density, soil texture, soil bulk density, soil erodibility, distance to streams, drainage density, elevation, slope gradient, aspect, local relief, plan curvature, profile curvature, topographic position index (TPI), topographic wetness index (TWI), steam power index (SPI), and land-use.

**Table 1.** Data layers of the spatial database for the predisposing factors.

| Predisposing Factor | Data Source | Format |
|---|---|---|
| Lithology | Geological Map of Calabria (Italy), 1:25,000 (CASMEZ 1967) | Vector (polygons) |
| Distance to faults Fault density | Derived from Geological Map of Calabria (Italy) 1:25,000 (CASMEZ 1967) and vector layer of ITHACA Catalogue (2019) (http://sgi1.isprambiente.it/geoportal/catalog/main/home.page, accessed on 10 December 2020) | Vector (lines) |
| Land use | Corine Land Cover map of Italy, scale 1:100,000 (ISPRA, 2018), (http://www.sinanet.isprambiente.it/it/sia-ispra/download-mais/corine-land-cover, accessed on 13 February 2021) | Vector (polygons) |
| Soil texture Soil bulk density | Soil map of Calabria (Italy), scale 1:250,000 (Calabria region, 2003) | Vector (polygons) |
| Soil erodibility | Soil erosion risk map of Calabria (Italy), scale 1:250,000—Calabria region, 2005 | Vector (polygons) |
| Distance to streams Drainage density | Derived from vector layer of drainage network of Calabria (Italy), Cartographic Center of Calabria region (http://geoportale.regione.calabria.it/opendata, accessed on 30 November 2020) | Vector (lines) |
| Elevation Local relief Slope gradient Slope aspect Plan curvature Profile curvature TPI TWI SPI | Derived from digital elevation model (DEM), with 5 m pixel size—Cartographic Center of Calabria region (http://geoportale.regione.calabria.it/opendata, accessed on 30 November 2020) | Raster |

The ancillary data used for the construction of the geolithological and fault lineaments maps were: the geological map of the Calabria region at scale 1:25,000, ITHACA Catalogue (available at http://sgi1.isprambiente.it/geoportal/catalog/main/home.page, accessed on 10 December 2020), photo interpretation and field surveys (Table 1). The lithotypes cropping out in the study area were grouped into 8 lithological classes as following (Figure 4a): alluvial deposits (Holocene), eluvial/colluvial deposits (Holocene), conglomerates and sands (Pliocene), sands and sandstones (Pliocene), silty clays (Pliocene), evaporitic limestones (Miocene), gneiss (Paleozoic), and granites (Paleozoic). The distance to faults was calculated by using the Euclidean distance (Figure 4b); whereas, the fault density map (Figure 4c) was elaborated by QGIS 3.16 software using the kernel density algorithm with a search area of 1 km$^2$.

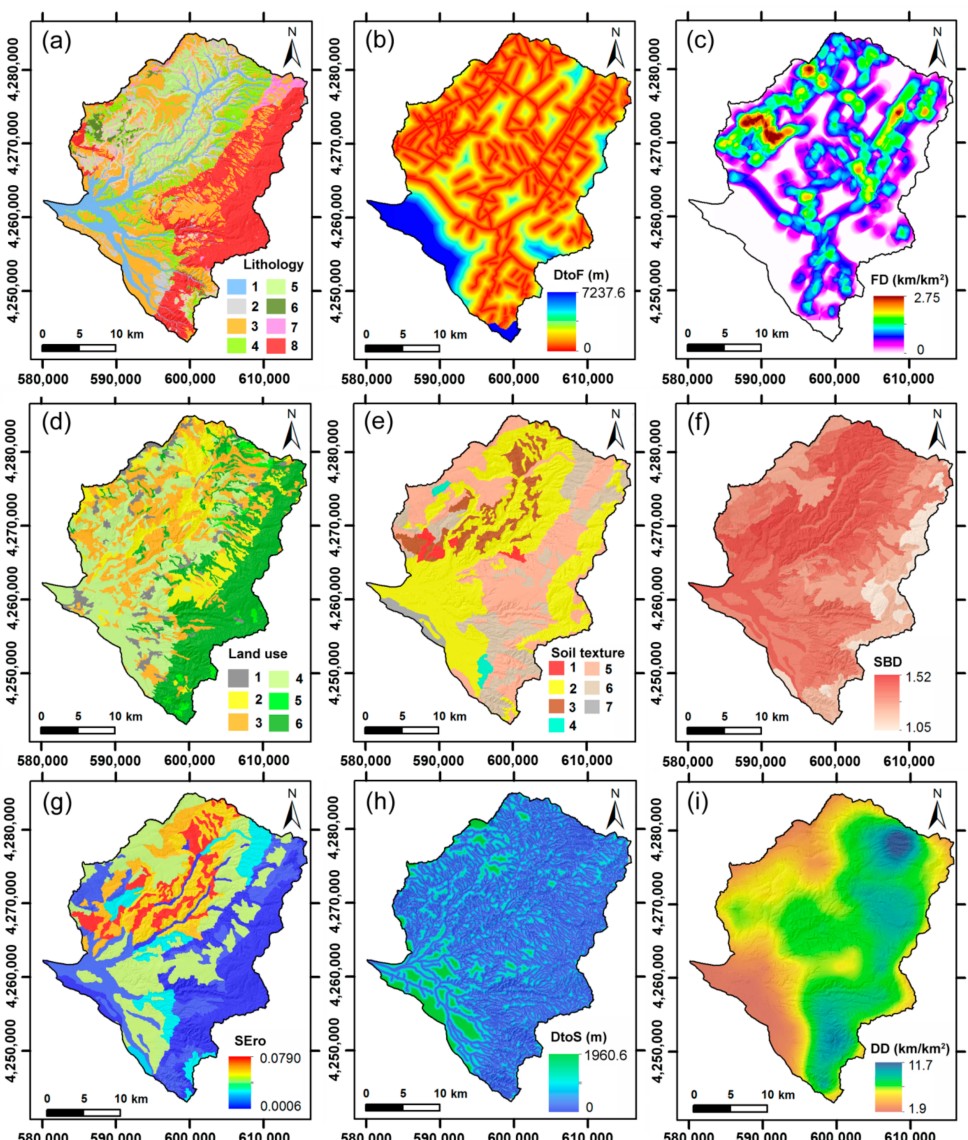

**Figure 4.** (**a**) Lithology map (Legend: 1 alluvial deposits—Holocene; 2 eluvial/colluvial deposits—Holocene; 3 silty clays—Pliocene; 4 conglomerates and sands—Pliocene; 5 sandstones—Pliocene; 6 evaporitic limestones—Miocene; 7 gneiss—Paleozoic; 8 granite—Paleozoic); (**b**) distance to fault map; (**c**) fault density map; (**d**) land use map (Legend: 1—artificial and/or urban areas; 2—arable areas; 3—heterogeneous agricultural areas; 4—fruit and olive grove areas; 5—scrub and/or herbaceous areas; 6—forest areas); (**e**) Soil texture map (Legend: 1—clay loam; 2—loam; 3—silty loam; 4—sandy clay loam; 5—sandy loam; 6—loamy sand; 7—sand); (**f**) soil bulk density map; (**g**) soil erodibility map; (**h**) distance to streams map; (**i**) drainage density map.

The land use map was extracted by the Corine Land Cover map of Italy (scale 1:100,000), downloaded from website of ISPRA (http://www.sinanet.isprambiente.it/it/sia-ispra/download-mais/corine-land-cover, accessed on 13 February 2021). In the study area land-use classes were detected as following: forest areas, scrub and/or herbaceous areas, fruit and olive grove areas, heterogeneous agricultural areas, arable areas, artificial, and/or urban areas (Figure 4d).

The predisposing factors related to soil features, such as soil texture and soil bulk density (Figure 4e,f), were obtained from the data enclosed in the digital soil map of Calabria (scale 1:250,000) [97]. Instead, the data concerning the soil erodibility factor (Figure 4g) are associated with the soil characteristics and include: texture, organic matter content, soil structure, and permeability. These data were extracted from soil erosion risk map of Calabria (scale 1:250,000) [104].

The drainage network data, available on the website of the Cartographic Center of Calabria region (http://geoportale.regione.calabria.it/opendata, accessed on 30 November 2020), were used to extract the following predisposing factors: distance to streams and drainage density (Figure 4h,i). In particular, the distance to streams was calculated by using the Euclidean distance; instead, the drainage density was defined considering the total length of stream network per unit area (1 km$^2$ in this study). This procedure was performed through the line density analysis tool in QGIS 3.16 software.

The topographic factors such as: elevation, slope gradient, aspect, plan curvature, profile curvature, TPI, TWI, and SPI were derived by a digital elevation model (DEM) with a spatial resolution of $5 \times 5$ m (Figure 5), using the SAGA-GIS software [105]. The used DEM was downloaded from the website of the Cartographic Center of Calabria region (http://geoportale.regione.calabria.it/opendata, accessed on 30 November 2020). The map of local relief (Figure 5d), which highlights the maximum difference in height for unit area, was obtained from DEM through the subtraction of the elevation maximum value from the minimum one within a moving window sized 1 km$^2$ [106,107].

All the data layers were converted into a raster format, with a resolution of $5 \times 5$ m pixel size, useful to the shallow landslide susceptibility analysis. A statistical description of the continuous predisposing factors is shown in Table 2.

**Table 2.** Descriptive statistics of the continuous predisposing factors.

| Predisposing Factors | Min | Max | Mean | S. Dev. |
|---|---|---|---|---|
| Distance to faults | 0.00 | 7237.60 | 808.80 | 997.60 |
| Fault density | 0.00 | 2.75 | 0.41 | 0.51 |
| Soil bulk density | 1.05 | 1.52 | 1.35 | 0.11 |
| Soil erodibility | 0.01 | 0.08 | 0.03 | 0.02 |
| Distance to streams | 0.00 | 1960.60 | 130.40 | 164.70 |
| Drainage density | 0.00 | 11.70 | 6.10 | 1.90 |
| Elevation | 0.00 | 1275.00 | 402.90 | 287.70 |
| Slope gradient | 0.00 | 78.20 | 14.10 | 11.10 |
| Local relief | 0.40 | 630.10 | 285.90 | 117.10 |
| Plan curvature | −0.27 | 0.19 | 0.00 | 0.01 |
| Profile curvature | −0.31 | 0.29 | 0.00 | 0.02 |
| TPI | −41.67 | 36.98 | 0.02 | 5.83 |
| TWI | 0.00 | 25.64 | 6.14 | 2.23 |
| SPI | 0.00 | 1965.40 | 5.82 | 9.75 |

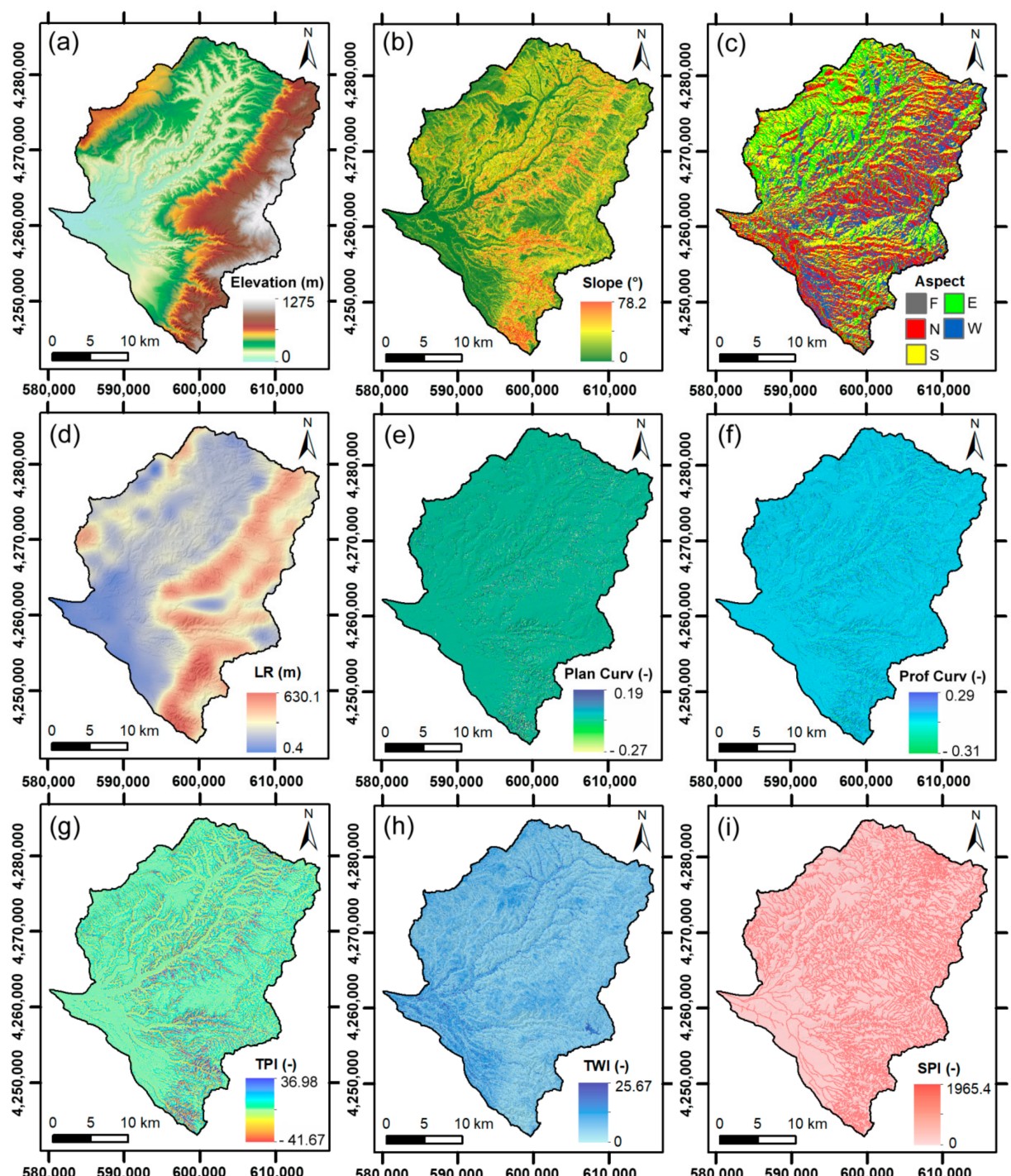

**Figure 5.** (**a**) Elevation map; (**b**) slope gradient map; (**c**) slope aspect map; (**d**) local relief map; (**e**) plan curvature map; (**f**) profile curvature map; (**g**) TPI map; (**h**) TWI map; (**i**) SPI map.

### 3.3. Shallow Landslide Susceptibility Modeling

### 3.3.1. Multi-Collinearity Analysis

Before to use the predisposing factors for shallow landslide susceptibility modeling (SLSM), a multi-collinearity analysis was performed both to exclude the highly correlated predisposing factors from landslide susceptibility analysis and to avoid any bias in model results. Multi-collinearity is defined as a condition in which two or more independent variables, in a multiple regression model, are linearly correlated. Multi-collinearity problem

may reduce the capability of model [65,108]. In order to detect the multi-collinearity among the 18 predisposing factors used in this study, the tolerance (TOL) and variance inflation factors (VIF) were calculated [71]. Values of VIF > 10 and TOL < 0.1 generally indicate serious problems of multi-collinearity [71]. Therefore, predisposing factors with VIF values more than 10 and TOL values less than 0.1 must be excluded from the susceptibility modeling [29,109]. The multi-collinearity analysis was carried out by the SPSS software.

### 3.3.2. Logistic Regression Analysis

Logistic regression (LR) is a generalized linear model widely used for mapping landslide susceptibility [18,29,42,54,110]. The aim of LR is to find the best model to describe the relationship between a dependent variable and a set of independent variables [109]. The dependent variable is dichotomous, whereas the independent variables can be continuous, or discrete, or a combination of the two types and they have not always a normal distribution. In this study, the dichotomous variables indicate the presence or absence of a landslide (1 landslide and 0 non-landslide), instead the independent variables represent the 18 landslide predisposing factors Figures 3 and 4. Therefore, the LR analysis allows to assess the spatial probability that landslide event occurring in an area [18,42]. The probability values range from 0 (no occurrence) to 1 (occurrence).

The LR function can be expressed as

$$P = \frac{1}{(1 + exp^{-z})} \tag{1}$$

$$z = a + b_1 x_1 + b_2 x_2 \cdots + b_n x_n \tag{2}$$

where $P$ is the probability of a landslide occurrence, $z$ represents the weighted linear combination of the independent variables, $a$ is the constant or intercept, $xi$ ($i$ = 1, 2, . . . n) are the independent variables and $b_i$ ($i$ = 1, 2, . . . n) are the regression coefficients of the independent variables.

To calculate the coefficients of each independent variable (predisposing factors) the maximum likelihood technique was used [111]. Subsequently, $z$ is transformed by means of Equation (1) into a logit variable (the natural log of the odds of presence or absence). Coefficients are expressed in log units; if a coefficient is positive, its transformed log value will be greater than one, meaning that the modeled event is more likely to occur. If a coefficient is negative, the odds of the event occurrence decrease; a zero (0) coefficient has a transformed log value of 1, meaning that this coefficient does not change the odds of the event one way or the other [112].

The LR model allows the integration of both continuous and categorical independent variables. In this study, continuous data such as: topographical predisposing factors, soil bulk density, soil erodibility, fault density, distance to faults, drainage density and distance to streams were used in LR analysis in their original format to avoid altering the state and information present in the predisposing factor maps. However, the categorized factors such as: lithology, soil texture, land use, and aspect were converted to continuous variables through the computation of the respective landslide frequency. Landslide frequency was expressed as the ratio between the number of shallow landslides and the area of the class of each predisposing factor (Table 3). Using landslide frequency to transform the nominal variables to numeric variables can avoid the creation of an excessively high number of dummy variables and it allows to evaluate the so-called 'previous knowledge' of landslide susceptibility [113].

**Table 3.** Categorical predisposing factors and related landslide frequency values used in the logistic regression analysis.

| Predisposing Factors | Class | Area (a) | Landslide Training Set (b) | Landslide Frequency (b/a) |
|---|---|---|---|---|
| | | km$^2$ | Count | Count/km$^2$ |
| Lithology | Alluvial deposits (Holocene) | 91.60 | 32 | 0.35 |
| | Eluvial/colluvial deposits (Holocene) | 27.32 | 16 | 0.59 |
| | Conglomerates and sands (Pliocene) | 199.01 | 192 | 0.96 |
| | Sandstones (Pliocene) | 95.08 | 171 | 1.80 |
| | Silty clays (Pliocene) | 153.68 | 279 | 1.82 |
| | Evaporitic limestones (Miocene) | 13.54 | 7 | 0.52 |
| | Gneiss (Paleozoic) | 12.33 | 33 | 2.68 |
| | Granite (Paleozoic) | 213.84 | 328 | 1.53 |
| Soil texture | Clay loam | 12.06 | 13 | 1.08 |
| | Loam | 360.77 | 290 | 0.80 |
| | Silty loam | 59.48 | 22 | 0.37 |
| | Sandy clay loam | 7.64 | 4 | 0.52 |
| | Sandy loam | 234.91 | 407 | 1.73 |
| | Loamy sand | 120.17 | 322 | 2.68 |
| | Sand | 11.38 | 0 | 0.00 |
| Land-use | Artificial and/or urban areas | 31.63 | 38 | 1.20 |
| | Arable areas | 110.20 | 84 | 0.76 |
| | Heterogeneous agricultural areas | 188.17 | 208 | 1.11 |
| | Fruit and olive grove areas | 224.08 | 220 | 0.98 |
| | Scrub and/or herbaceous areas | 15.97 | 62 | 3.88 |
| | Forest areas | 236.36 | 446 | 1.89 |
| Slope aspect | Flat | 1.08 | 0 | 0.00 |
| | North | 214.74 | 231 | 1.08 |
| | East | 137.61 | 203 | 1.48 |
| | South | 209.16 | 410 | 1.96 |
| | West | 243.61 | 214 | 0.88 |

By performing LR analysis on the training dataset, using stepwise logistic regression model implemented in SPSS 26.0 software, the regression coefficient for each predisposing factor was calculated. The model fitting to the observed data was evaluated by computing the values of the Cox and Snell and Nagelkerke pseudo-$R^2$ in addition to the statistic$-2LL$.

The logistic regression component of the software SPSS also provides the results of the model chi-square test, which allows to assess the global significance of the regression coefficients. The significance was also evaluated individually for each independent variable incorporated in the model by means of the Wald test [112]. Afterwards, the regression coefficients of the predisposing factors were imported in GIS and by using Equations (1) and (2), the shallow landslide susceptibility map was performed.

There are different methods of classifying the susceptibility values, which are integrate into the GIS software. Hence, in the present study, four classification systems (i.e., natural breaks, equal interval, quantile, and geometric interval) were tested to draw the best shallow landslide susceptibility map. This comparative evaluation allowed us to identify which susceptibility maps are similar and which best define the shallow landslide susceptibility distribution in the study area. Subsequently, the map that best matched the information was chosen.

3.3.3. Validation of Susceptibility Model

The model validation phase constitutes one of the most important issues in landslide susceptibility analysis [114]. In this study, the goodness-of-fit (assessment of the model accuracy based on the training subset) and predictive skill (model accuracy based on the validation subset) of the SLSM were evaluated through the following elements: confusion

matrix and related statistic indexes, receiver operating characteristic (ROC) curve, area under curve (AUC), and the kappa coefficient [114–116]. The confusion matrix shows the amount of True Positives (*TP*), i.e., pixels predicted unstable and observed unstable (areas with landslides), True Negatives (*TN*), i.e., pixels predicted stable and observed stable (areas without landslides), False Positives (*FP*), i.e., pixels predicted unstable but observed stable and False Negatives (*FN*), i.e., pixels predicted stable but observed unstable.

The ROC curve measures the goodness-of-fit and the prediction performance of the model [115,116] through plotting, for different susceptibility threshold values, the *TP* rate (Sensitivity) and the *FP* rate (1-Specificity). The sensitivity and specificity were calucaleted using the equations

$$Sensitivity = \frac{TP}{TP + FN} \tag{3}$$

$$Specificity = \frac{TN}{FP + TN} \tag{4}$$

The AUC varies from 0.5 (diagonal line) to 1, with higher values indicating a better predictive capability of the model. Hosmer and Lemeshow [112], on the basis of the computed AUC values, classified the predictive performance as: acceptable for AUC $\geq$ 0.7, excellent for AUC $\geq$ 0.8, and outstanding for AUC $\geq$ 0.9. ROC curves were drawn both for the validation and training datasets, in order to evaluate predictive performances of the models and to further investigate their fit to the training observations.

The kappa coefficient was calculated using the components of the confusion matrix [117]. The kappa coefficient (*K*) was used to evaluate the reliability of the model to classify the landslide pixels [109]. It was calculated as the proportion of the observed agreement beyond that occurring by chance, using the equation

$$K = \frac{P_{obs} - P_{exp}}{1 - P_{exp}} \tag{5}$$

where $P_{obs}$ is the proportion of pixels classified correctly as landslide or non-landslide and $P_{exp}$ is the proportion of pixels for which the agreement is expected by chance [118]. $P_{obs}$ and $P_{exp}$ were calculated as

$$P_{obs} = \frac{TP + TN}{n} \tag{6}$$

$$P_{exp} = \frac{(TN + FP) \times (TN + FN) + (FN + TP) \times (FP + TP)}{\sqrt{N}} \tag{7}$$

where *TP* (true positive) and *TN* (true negative) are the numbers of pixels classified correctly as landslide occurrence and no landslide occurrence, respectively; *FP* (false positive) and *FN* (false negative) are the numbers of pixels classified incorrectly in the confusion matrix; *n* is the proportion of pixels classified correctly as landslide or not landslide; N is the total number of pixels. In agrement to Landis and Koch [119] the performance of a model based on the kappa coefficient can be classified as: $\leq$0 = poor, 0–0.2 = slight, 0.2–0.4 = fair, 0.4–0.6 = moderate, 0.6–0.8 = substantial, and 0.8–1 = almost perfect.

### 3.3.4. Relative Importance of Predisposing Factors

A sensitivity analysis was performed using jackknife-based test to detect the contribute of each predisposing factor in the shallow landslide susceptibility model [57,120,121]. Sensitivity analysis plays an important role in modeling process, because it helps to understand how model results are affected by input data. In the jackknife test the importance of a certain predisposing factor is estimated by excluding each factor in turns in the model run constructed and using the remaining factors. The accuracy of the models were performed by means the computation of AUC value [58,120]. The sensitivity analysis, performed using the training set data, allowed to identify the relative importance (*RI*) of each predisposing factor in the implementation of the landslide susceptibility model, which includes

all predisposing factor [120,122]. To compute the *RI* of each predisposing factor, used in the susceptibility model, the following equation was employed [58]

$$RI = \frac{(AUC_{all} - AUC_i)}{AUC_{all}} \times 100 \tag{8}$$

where $AUC_{all}$ is the value of *AUC* computed from the prediction model using all predisposing factors, while $AUC_i$ represents the *AUC* value when *i*th factor was excluded from the computation of model. Therefore, to higher *RI* value corresponds a greater influence of the excluded factor in the susceptibility model [122].

## 4. Results

### 4.1. Detection of Multi-Collinearity between Predisposing Factors

One of the important steps in the landslide susceptibility model is the selection of the predisposing factors and to identify which are redundant with respect to others [75]. The outcomes of the multi-collinearity test showed that there are not significant correlations among predisposing factors in the study area, consequently all predisposing factors were used to building SLSM. The results of multi-collinearity analysis are summarized in Table 4. The values of TOL and VIF indices for all 18 predisposing factors are >0.1 and <10, respectively. These results show absence of multi-collinearity problems among them. The VIF values of the selected predisposing factors range from 1.058 to 3.144, whereas for the TOL index, the values are included between 0.318 and 0.945 (Table 4).

**Table 4.** Results of the multi-collinearity analysis between the predisposing factors.

| Predisposing Factors | Multi-Collinearity | |
| --- | --- | --- |
| | **TOL** | **VIF** |
| Lithology | 0.732 | 1.366 |
| Distance to faults | 0.778 | 1.286 |
| Fault density | 0.673 | 1.486 |
| Land-use | 0.637 | 1.569 |
| Soil texture | 0.567 | 1.763 |
| Soil bulk density | 0.462 | 2.164 |
| Soil erodibility | 0.570 | 1.755 |
| Distance to streams | 0.659 | 1.518 |
| Drainage density | 0.454 | 2.204 |
| Elevation | 0.318 | 3.144 |
| Slope gradient | 0.346 | 2.890 |
| Slope aspect | 0.920 | 1.087 |
| Local relief | 0.323 | 3.097 |
| Plan curvature | 0.717 | 1.395 |
| Profile curvature | 0.722 | 1.384 |
| TPI | 0.734 | 1.362 |
| TWI | 0.451 | 2.218 |
| SPI | 0.945 | 1.058 |

### 4.2. Assessment of Shallow Landslide Susceptibility

The SLSM, within the study area, was computed through LR considering the 18 predisposing factors (lithology, distance to faults, fault density, land use, soil texture, soil bulk density, soil erodibility, distance to streams, drainage density, elevation, slope gradient, slope aspect, local relief, plan curvature, profile curvature, TPI, TWI, and SPI) and the training dataset, consisting of the 70% of the landslides and no-landslides. At this regard, the SPSS software package was employed. The SLSM was performed by applying the forward stepwise logistic regression algorithm. The result achieved through Hosmer and Lemeshow test highlighted that the goodness of fit of the LR model is acceptable because the significance of Chi-square is larger than 0.05 (Table 5). In addition, the values of Cox

and Snell R$^2$ (0.575) and Nagelkerke R$^2$ (0.767) confirmed that the LR model performed very well.

**Table 5.** Overall statistics of the logistic regression model.

| Hosmer and Lemeshow Test | | | −2 Log Likelihood | Cox and Snell R$^2$ | Nagelkerke R$^2$ |
|---|---|---|---|---|---|
| Chi-square | *df* | Sig. | | | |
| 149.822 | 8 | 0.137 | 1122.742 | 0.575 | 0.767 |

　　　The regression and standardized coefficients obtained by LR analysis are listed in Table 6. The sign of the values of regression coefficient indicates if the predisposing factor is positively or negatively correlated to shallow landslide occurrence. These coefficients were then used to produce the shallow landslide susceptibility map of the study area. This map was constructed implementing Equation (1) in GIS software. For each pixel of the study area, the values of the predisposing factors were multiplied with their respective regression coefficients (*bi*), then they were summed up and added to the estimated constant (*a*), so the susceptibility of shallow landslide was estimated. The values of susceptibility range between 0.001 and 0.988, with a mean value of 0.333 and a standard deviation of 0.327. The pixels with susceptibility values closer to 1 denotes a high probability of shallow landslides occurrence, whereas the pixels with values closer to 0 indicate a low probability of shallow landslide occurrence.

**Table 6.** Coefficients and test statistics of the predisposing factors used in the logistic regression model.

| Predisposing Factor | *b* | S.E. | Wald | df | Sig. | Exp(*b*) |
|---|---|---|---|---|---|---|
| Lithology | 0.462 | 0.127 | 13.196 | 1 | 0.000 | 1.588 |
| Distance to faults | 0.208 | 0.202 | 1.051 | 1 | 0.031 | 1.231 |
| Fault density | 0.441 | 0.149 | 8.747 | 1 | 0.003 | 1.555 |
| Land-use | 0.061 | 0.155 | 0.155 | 1 | 0.044 | 1.063 |
| Soil texture | 0.589 | 0.136 | 18.686 | 1 | 0.000 | 1.803 |
| Soil bulk density | −3.781 | 1.217 | 9.652 | 1 | 0.002 | 0.023 |
| Soil erodibility | 1.532 | 0.507 | 9.092 | 1 | 0.003 | 4.626 |
| Distance to streams | 0.347 | 0.135 | 6.611 | 1 | 0.010 | 1.415 |
| Drainage density | −0.042 | 0.057 | 0.544 | 1 | 0.046 | 0.959 |
| Elevation | −0.003 | 0.001 | 21.370 | 1 | 0.000 | 0.997 |
| Slope gradient | 0.167 | 0.011 | 248.296 | 1 | 0.000 | 1.182 |
| Slope aspect | 0.352 | 0.182 | 3.756 | 1 | 0.026 | 1.422 |
| Local relief | 0.005 | 0.001 | 26.061 | 1 | 0.000 | 1.005 |
| Plan curvature | −6.918 | 1.685 | 1.019 | 1 | 0.033 | 0.001 |
| Profile curvature | 1.918 | 0.508 | 14.258 | 1 | 0.000 | 6.808 |
| TPI | −0.024 | 0.015 | 2.486 | 1 | 0.011 | 0.977 |
| TWI | −0.011 | 0.063 | 0.034 | 1 | 0.009 | 0.989 |
| SPI | 0.007 | 0.010 | 0.432 | 1 | 0.005 | 1.007 |
| Constant (*a*) | −1.262 | 0.445 | 0.506 | 1 | 0.005 | 0.283 |

*b* = coefficient; S.E. = Standard error of estimate; Wald = Wald chi-square values; df = degree of freedom; Sig. = significance; Exp(*b*) = exponential coefficient.

　　　Thereafter, the shallow landslide probability values, computed for each pixel, were classified into five classes of susceptibility (i.e., very low, low, moderate, high, and very high). To define the best method to categorize the susceptibility values, a comparative analysis of four classification methods (natural break, equal interval, quantile, and geometric interval) was carried out. Consequently, the map that best matched the informations in the study area was chosen. The Pearson correlation coefficient was used as a first evaluation of the statistical significance of the difference in the susceptibility classification between each pair of methods. Table 7 shows that all classification methods have high correlations, the

correlation values ranges from 0.79 to 0.96. The lowest correlation was obtained by comparison between equal interval and quantile methods; whereas, the highest correlation was detected by comparison of the maps obtained by quantile and geometric interval. Overall, the high values of correlation, obtained by the four classification methods, highlights a good similarity among them to classify the susceptibility values.

**Table 7.** Pearson's correlation among the susceptibility maps obtained by using four different classification methods.

| Classification Methods | Natural Break | Equal Interval | Quantile | Geometric Interval |
|---|---|---|---|---|
| Natural break | 1.00 | 0.95 | 0.86 | 0.88 |
| Equal interval | - | 1.00 | 0.79 | 0.81 |
| Quantile | - | - | 1.00 | 0.96 |
| Geometric interval | - | - | - | 1.00 |

Significant at $p < 0.01$.

The percentage values of shallow landslide susceptibility classes obtained by the four classification methods are summarized in Figure 6. The cross-comparisons of the susceptibility maps, in terms of the coverage differences among the recognized classes (Figure 6), indicate a marked difference of the employed classification methods. In particular, considerable differences were observed for the areal distribution mapped as very low susceptibility; whereas, the zones mapped with very high susceptibility are more similar. The outputs from the overlay of the susceptibility maps with the shallow landslide training and validation datasets are presented in Figure 6 as well. For the training dataset, the four maps generally display a gradual increase in the percentage of landslide cells ranging from 'very low' to 'very high' susceptibility class. However, the shallow landslides of both training and validation sets fall in the very high susceptibility class (Figure 6). In addition, landslide density was calculated to evaluate the best classification method to draw the shallow landslide susceptibility map of the study area [123,124]. The landslide density was computed using the ratio between the percentages of total shallow landslides and the percentages of each susceptible class. Obviously, greater susceptible classes should have higher values of landslide density in reliable landslide susceptibility maps [124]. Overall, the achieved outcomes showed lower values of landslide density in the very low and low susceptibility classes, and high landslide density in the very high susceptibility class for all the classification method (Figure 6). Therefore, the results indicate that the natural break is the most robust method for the classification of susceptibility map, because it showed the highest occurrence both of shallow landslides and of highest value of landslide density in the very high susceptibility class (Figure 6). Consequently, the final shallow landslide susceptibility map was realized according to the selected classification method (Figure 7a).

The relative distribution of the shallow landslide susceptibility classes is show in Figure 6. The data display that most of the study area was predicted with a very low (44.2%) or low (17.5%) susceptibility to shallow landslide; whereas, a minor area was classified as having high (11.5%) or very high (14.5%) susceptibility (Figure 6). The visual analysis of the map shows that high and very high susceptible areas are located mainly along the left side of the Mesima basin; instead, the valley floor is characterized by susceptibility values ranging from very low to low (Figure 7a).

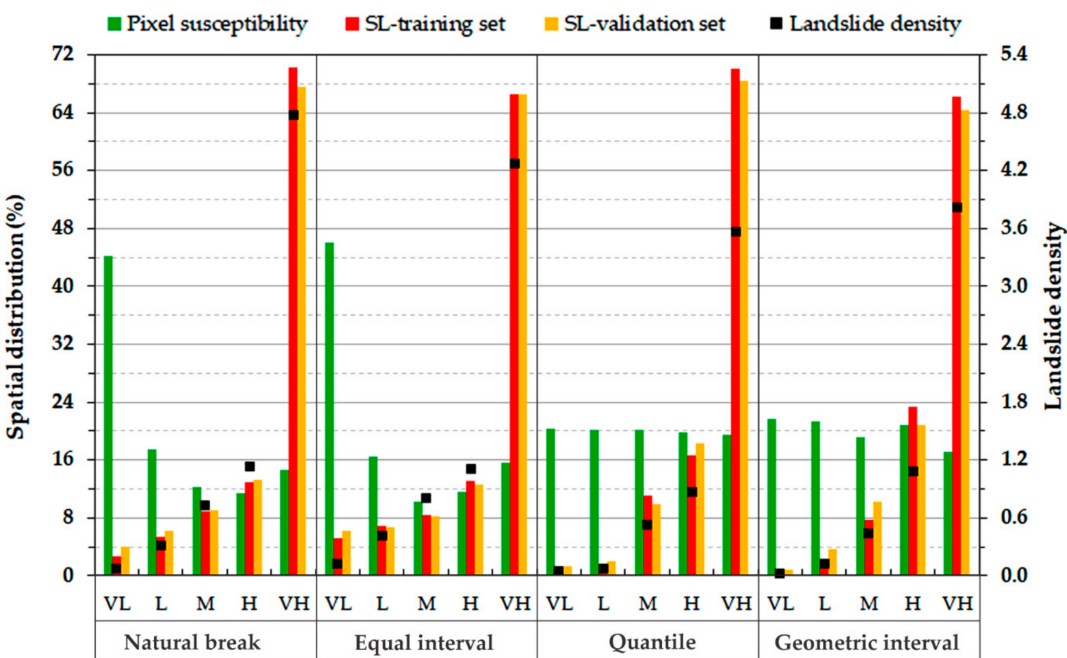

**Figure 6.** Comparison of the classification methods tested to select the best classification method for draw the shallow landslide susceptibility map.

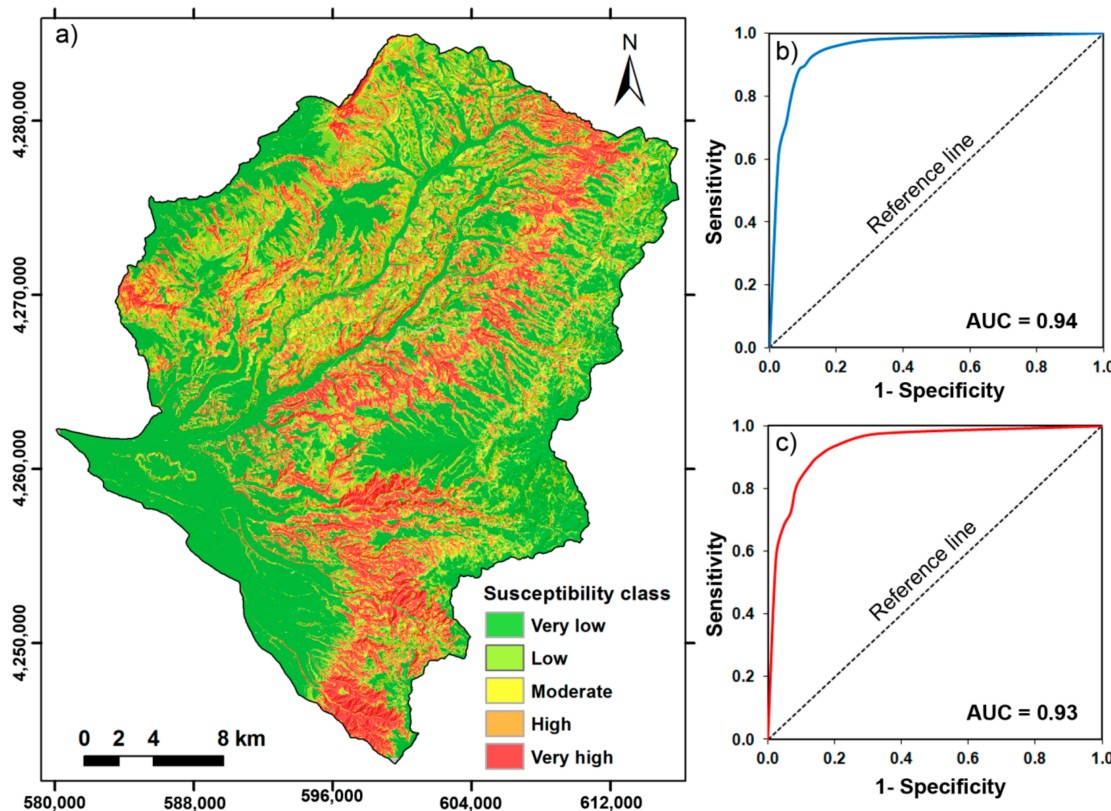

**Figure 7.** (**a**) Shallow landslide susceptibility map of the study area, obtained by the natural break classification method; (**b**) ROC curve and AUC value of training set; (**c**) ROC curve and AUC value of validation set.

The overlapping between susceptibility map and shallow landslides inventory allowed to verify the spatial distribution of the shallow landslides in the five susceptibility classes (Figure 6). The results showed that most of the shallow landslides (82.4%) occur in

areas with high and very high susceptibility classes; whereas, only the 8.7% of the shallow landslides occurred in low and very low susceptibility classes. The remaining 8.9% of the considered shallow landslides fall in the moderate susceptibility class.

### 4.3. Performance of the Model and Accuracy of the Susceptibility Map

The validation of SLSM and related susceptibility map were done with the help of several statistic indexes and ROC curve analysis [114,115,117], using training and validation datasets. The statistical indices of the training dataset give useful information of the reliability of the SLSM; instead, the statistical indices of the validation dataset provide well indication of the predictive capabilities of the SLSM [125]. The results of training phase showed that the LR model, with a cut-off of 0.5, classifies correctly the 90.5% of shallow landslides (Sensitivity index) and the 88.9% of no-shallow landslide (specificity index). The overall predicted accuracy was equal to 89.7% (Figure 8a), showing a high degree of fit of the susceptibility model with the training dataset. The value of Kappa coefficient, calculated for the training dataset, was equal to 0.79, denoting a substantial agreement between the observed and predicted shallow landslides [119].

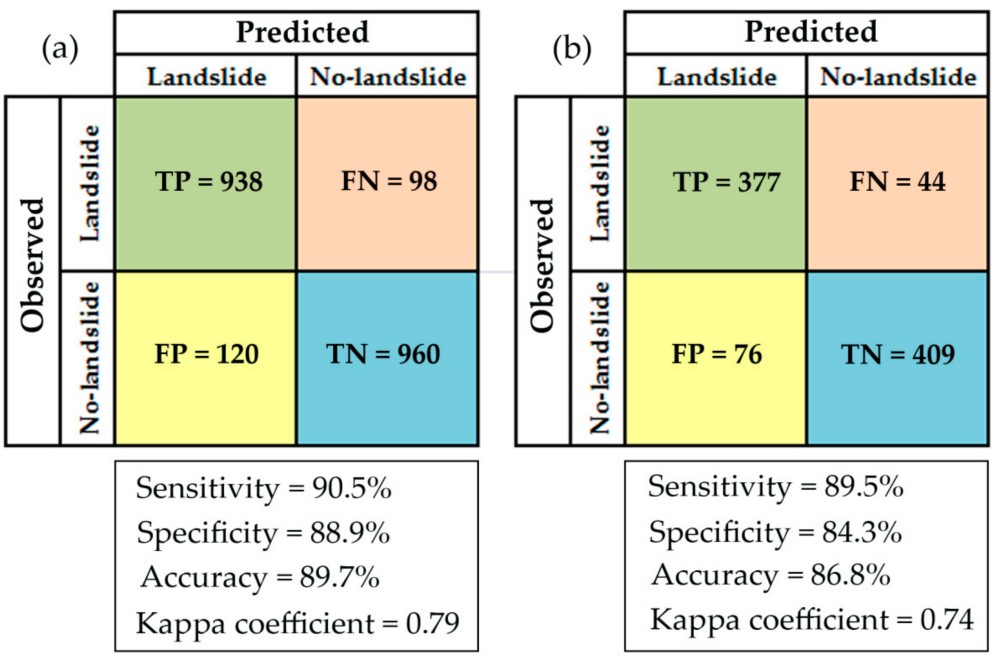

**Figure 8.** Confusion matrix and statistical indexes analysis of shallow landslide susceptibility model using training dataset (**a**) and validation dataset (**b**) TP = true positives, TN = true negatives, FP = false positives, and FN = false negatives for training and validation datasets.

The results of the prediction capability of the SLSM are summarized in Figure 8b. It could be observed that the accuracy of the LR model was 87%, highlighting a high prediction results. Sensitivity of the model was 89.5%, indicating that the amount of the shallow landslides, accurately predicted, is 89.5%; whereas, the value of specificity was 84.3%, which indicates that the amount of the no-shallow landslide pixels accurately predicted is 84.3%. The Kappa coefficient is 0.74, showing a high performance of the SLSM (Figure 8b).

Determination of the accuracy of the SLSM and related susceptibility map were also achieved by plotting ROC curves and calculating their AUC values. The ROC curve was plotted both for the training and validation datasets (Figure 7b,c). The ROC curve of the training dataset indicates the success rate of the model and the reliability of the susceptibility map; whereas, the ROC curve of the validation dataset estimates the prediction rate and the accuracy of the susceptibility map. The analysis of ROC curve (Figure 7b,c) displayed

an excellent performance of the SLSM both for the training and test datasets. In particular, at a confidence level of 95%, the ROC curve showed an AUC value of 0.94 for the training set and an AUC of 0.93 for the validation set (Table 8).

**Table 8.** Results of the area under ROC curve (AUC) analysis.

| | Area | Standard Error | Asymptotic Significance | Asymptotic 95% Confidence Interval | |
|---|---|---|---|---|---|
| | | | | Lower Bound | Upper Bound |
| Training set | 0.940 | 0.005 | 0.000 | 0.930 | 0.951 |
| Validation set | 0.930 | 0.009 | 0.000 | 0.913 | 0.947 |

*4.4. Analysis of the Relative Importance of Predisposing Factors*

Utilizing Jackknife test coupled to computation of *RI* values of each predisposing factors yielded information on the different contributions of the predisposing factors in the SLSM. The results of the assessment of *RI* of the predisposing factors, calculated using the Equation (8), were showed in Figure 9. All the predisposing factors have positively contributed for the assessment of the shallow landslides susceptibility in the study area. The predisposing factors with higher *RI* values indicate high predictive capability in shallow landslide susceptibility model and vice versa. The analysis of *RI* showed that the slope gradient, with a *RI* of 9.6%, was the most important factor to control the spatial distribution of shallow landslides. The second most important factor is TWI (RI = 8.5%), followed by soil texture (RI = 7.1%) and lithology (RI = 6.3%). The smallest values of RI, among the 18 predisposing factors, were observed for fault density and SPI, with *RI* values equal to 1.5% and 1.1%, respectively (Figure 9). The achieved results demonstrate that all predisposing factors contributed for the shallow landslide susceptibility modeling in the study area.

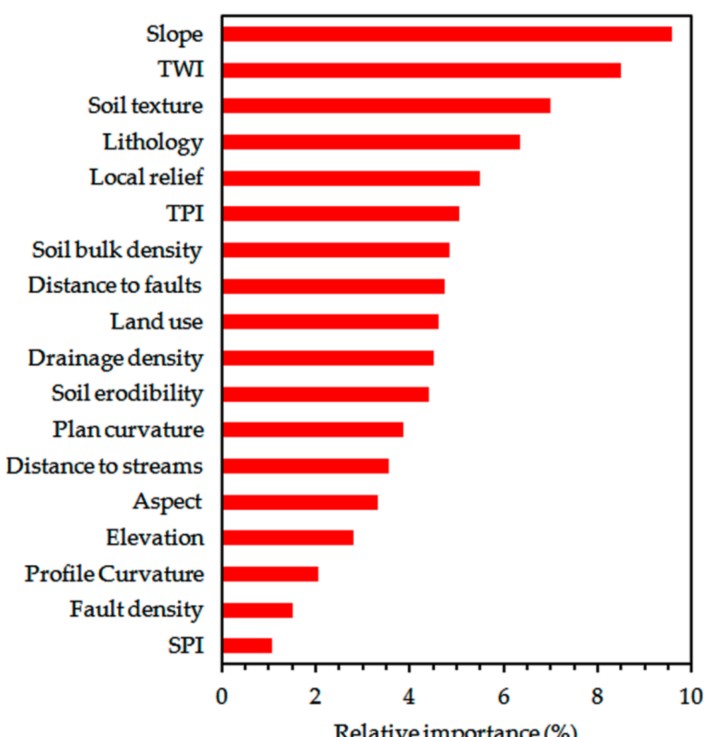

**Figure 9.** Relative importance of different predisposing factors used in the shallow landslide susceptibility model.



## 5. Discussion

An accurate landslide susceptibility mapping is an essential task to evaluate the landslide hazard, the management and mitigation of risk. The reliability of landslide susceptibility maps depends both by the quality of available training data and the applied model [24,25,27,69,102,124]. Training data may also be affected by distribution, size, and resolution data. In the last two decades, several approaches for assessing landslide susceptibility were used [29–32,44,57,102,126]. Susceptibility models are mainly driven by the data given to the algorithm. The choice of which geo-environmental variables can be used to evaluate the landslide susceptibility is a critical step in any susceptibility modeling [40,69,102,127]. For this purpose, a set of 18 predisposing factors such as: lithology, distance to faults, fault density, land-use, soil texture, soil bulk density, soil erodibility, distance to streams, drainage density, elevation, slope gradient, slope aspect, local relief, plan curvature, profile curvature, TPI, TWI, and SPI were selected on the basis both of previous scientific knowledge and new investigations in the study area [18,22,41,56,69,78,102]. However, the interdependencies of the previously mentioned factors, by applying multi-collinearity analysis, were tested. The obtained results showed no multi-collinearity among the predisposing factors, so all factors were used in the susceptibility analysis.

Based on the shallow landslide inventory map, the 70% of the total recognized shallow landslides were employed to generate the SLSM and the remaining 30% was used for validation purpose.

The SLSM results showed high accuracy, indeed during the training process the confusion matrix display a percentage correct of 89.7% and 86.8% is the one for the validation process (Figure 7). The examination of the results shows that the SLSM has high values both of sensitivity and of specificity. These values testify that the used model identify very well both the areas affected by shallow landslides and the ones not affected by shallow landslides. These results may be linked to the correlation between the nature of the model and the choices of important analysis parameters such as: the number, type, and quality of predisposing factors, as well as the size of training and validation datasets.

The overall results of the validation process, via ROC analysis, show that the SLSM supplies excellent AUC values (>0.90), both for the training and the validation datasets, coupled with high values of kappa coefficient (Figure 7) denote that LR is reliable and robust method for predicting susceptibility of shallow landslide in the study area. The LR model, due to flexible and nonlinear characteristics, is able to deal with a wide range of data, which are non-symmetrical and present complex input–output relationships, typical in the natural environment [109,128]. This outcome is comparable with recent studies concerning the high performance of the LR method, its strong capability in handling large datasets and its efficiency in non-exclusion any predisposing factor during computation phase of susceptibility [18,29,42,45,128].

The selection of the predisposing factors and their importance values have an impact on the accuracy of susceptibility zoning of an area. Therefore, many researchers tried to underline the importance to evaluate the contribution of each predisposing factor on the distribution of landslides [25,45,70,71,77,103,109,129]. In this study, the influence of each predisposing factor, used for shallow landslide susceptibility modeling, was tested by means jackknife test and *RI* index. The findings suggest that all predisposing factors contributed positively to implement the SLSM (Figure 8). However, the *RI* index showed that each predisposing factor had different importance in the susceptibility model construction. The greatest contribution to shallow landslide occurrence comes from slope gradient, which has a *RI* value of approximately 9.6% (Figure 8). Other important factors were: TWI (8.5%), soil texture (7.1%), lithology (6.4%), and local relief (5.5%). The achieved results support the previously results obtained in other studies, which showed that the shallow landslide distribution is highly controlled by morphometric factors and lithology features [21,25,73,75,103,129–133]. The key role played by slope gradient is related to its controls on the shear stress of the soil or unconsolidated material, which generally increases with the rise of slope angle [58,78,133]. The TWI represents an effective hydrological factor

contributing on shallow landslide susceptibility as well [25]. The TWI helps to distinguish the areas affected by an increased soil moisture content and water accumulation [56,63]. In our research, the areas of the watershed identified by values of TWI mid-high (8–20) are more exposed to shallow landslide occurrence.

Regarding the comparison between shallow landslide distribution and soil texture, the landslide frequency values are listed in Table 2. The results showed that shallow landslides mainly occurring on soils constituted by high sand fraction, such as: loamy sand and sandy loam texture. These soils are more prone to water infiltration during rainfalls and consequently the probability of shallow landslide occurrence is high [134].

Lithology is considered one of the most important factors in landslide susceptibility assessments, because it directly affects geomechanical and hydraulic properties of bedrock and of soil cover [70,135]. The highest values of landslide frequency were achieved in granitic rocks, gneissic rocks, and silty clays deposits, which are the most prone to shallow landslides, as summarized in Table 3. The study area is characterized by widespread granitic rocks affected by intense weathering processes that significantly reduce the rock strength and facilitate shallow slope failures. These results are in agreement with previous studies carry out in several areas of Calabria region [9,16,19,50,78,136].

Contrary, the alluvial and eluvial/colluvial deposits, typically occupies flat stable areas and thus it is not prone to shallow landslide process.

Four classification methods (natural break, equal interval, quantile, and geometric interval) were used to divide susceptibility map into five classes: very low, low, medium, high, and very high. The comparison of the four classification methods enabled to determine the best classification system in terms both of shallow landslide distribution and of landslide density within each susceptibility class. The Pearson correlation analysis highlights a good similarity of the four methods to classify the susceptibility values (Table 7). Indeed, the obtained four maps show similar identification and classification of the spatial distribution of the shallow landslide susceptibility (Figure 6). The quantile and geometric interval methods overestimated the high and very high susceptibility classes of the study area. While the spatial pattern of the equal interval and natural break maps was very similar. The natural break method resulted more consistent predicting potential landslides more efficiently and reliably in the study area. Therefore, the natural break classification method was selected to output the susceptibility map (Figure 7a). According to this map, the spatial distribution of the susceptibility classes denoted that high values of susceptibility were mainly observed along the left side of the Mesima basin. The area is characterized by a mountain landscape, highly dissected by V-shaped valleys with steep slopes. Granitic rocks, highly fractured and weathered, frequently mantled by thick regolith materials, both prone to shallow landslides, dominate in this area [9,10,78]. In addition, several areas characterized by slopes carved in silty clays lithologies and covered by soils with silty loam texture, both highly erodible, recorded high and very high shallow landslide susceptibility values [136]. Instead, low and very low shallow landslide susceptibility values affect more than 61% of the study area, which is mostly dominated by slope gradient ranging between 0 and 15 degrees (flat areas or gently slopes) and constituted by eluvial/colluvial deposits and/or alluvial deposit.

The comparison between susceptibility map and shallow landslide inventory map shows that about 90.9% of the observed shallow landslides occurred in the areas with susceptibility values ranging from moderate to very high (Table 6). The founded high percentage confirms the effectiveness of the susceptibility map. Moreover, comparing Figure 7a to Figure 3a, it was possible calculated that more than 85% of the no-shallow landslide points (for training and validation sets) are contained in the area classified as very low and low susceptible.

Overall, the achieved goals testify the reliability and effectiveness of the proposed model for determining the spatial probability of the future occurrence of shallow landslides in this sector of the Calabria region. A comparison between our results and other studies is difficult due to different sampling strategies for training and validation datasets parti-

tioning, different spatial scales, different sets of predisposing factors. However, taking into consideration the validation results, in terms of ROC curve and AUC values, our results are consistent with several previous studies concerning the use of LR model for mapping the landslide susceptibility, e.g., [18,47,51,72,110,131,137]. Moreover, the used approach highlights the roles of predisposing factors in the assessment of the shallow landslide susceptibility, giving useful results for researchers, decision makers, and land planners.

Finally, some assumptions/limitations of this study have to be pointed out. Considering the time-scale of the analysis and the landslide typology, the landslide inventory does not include the total number of recorded landslide events within the study area. Selecting the independent variables to be considered in the model and ranking their importance can be an effective tool for narrowing down the list of potential predisposing factors to be included in susceptibility modeling. Concerning the LR model, the choices for the analysis of independent variables prove to be critical and play a major role in the relative accuracy of the outcomes. The examination of alternative choices for these independent variables could lead to different results. Moreover, the applied model estimates the mean degree of impact of predisposing factors, which may differ locally in different parts of the study area. In fact, they represent the relation between landslide occurrence and causal factors for the whole study area without considering the spatial non-stationarity. Further studies are also to be focused on the exploration of this non-stationarity through the implementation of "local" models like geographically weighted regression [138].

## 6. Conclusions

The prediction of the shallow landslide susceptibility is one of the most important tasks in landslide hazard and risk determination. The reliability of a susceptibility model depends on the available and accuracy of the data and on the used method.

In this study, LR method was tested for a shallow landslide susceptibility modeling in Mesima basin, Calabria region (South Italy). Eighteen predisposing factors were considered, as follow: lithology, distance to faults, fault density, land-use, soil texture, soil bulk density, soil erodibility, distance to streams, drainage density, elevation, slope gradient, slope aspect, local relief, plan curvature, profile curvature, TPI, TWI, and SPI.

The performance of the SLSM model was quantitatively evaluated through the ROC curve analysis and Kappa coefficient. The outcomes showed that LR method gave satisfactory performance to predict the spatial occurrence of shallow landslides (AUC and K coefficient in the validation dataset are 0.93 and 0.74 respectively).

Regarding the four classification systems (i.e., natural breaks, equal interval, quantile, and geometric interval) tested, the natural break is the best method to draw the susceptibility map of the study area. The obtained susceptibility map is constituted by five classes: very low, low, moderate, high, and very high susceptibility, showing that high and very high susceptibility classes enclose the 26% of the total area. Comparing susceptibility map and shallow landslide inventory map, a satisfactory spatial agreement between the different classes of susceptibility and shallow landslide distribution was obtained. The 83.1% of the shallow landslides of the training set and 80.8% of the shallow landslides of validation set were correctly classified, falling in high and very high susceptibility classes.

The results highlighted also that the selection and the incidence evaluation of the predisposing factors is a key requirement for a valid modeling of landslide susceptibility. According to the results of *RI* index, it can be observed that all used predisposing factors had significant impact on the LR performance. However, the predisposing factors that more influenced the SLSM were: slope gradient, TWI, soil texture, and lithology.

Overall, the high reliability of the susceptibility map contributes to delineate in detail the areas with a high probability of future shallow landslides in Mesima basin. Thus, the achieved goals increase the knowledge of spatial prediction of shallow landslides in a high risk area of the Calabria region, giving a valuable tool that will can be used by local, regional and national authorities both for land-use planning as well as for management and mitigation of the shallow landslide risk.

However, in the future, other statistical methods and ensemble of machine learning techniques for landslide susceptibility modeling will be tested in this area.

**Author Contributions:** Conceptualization, M.C. and F.I.; Analysis of geological and geomorphological features of the study area and investigating shallow landslides, M.C. and F.I.; Formal analysis, and data curation M.C.; Writing—original draft, M.C. and F.I.; Writing—review and editing, M.C. and F.I. Both authors have read and agreed to the published version of the manuscript.

**Funding:** This research received no external funding.

**Institutional Review Board Statement:** Not applicable.

**Informed Consent Statement:** Not applicable.

**Data Availability Statement:** The data presented in this study are available on request.

**Acknowledgments:** This work is also part of the Progetto DTA.AD003.487 "TIPIZZAN-Tipizzazione di frane su area vasta e su singolo versante" of the CNR—Department of "Scienze del sistema Terra e Tecnologie per l'Ambiente".

**Conflicts of Interest:** The authors declare no conflict of interest.

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
