# Peer review of "Modeling Shallow Landslide Susceptibility and Assessment of the Relative Importance of Predisposing Factors, through a GIS-Based Statistical Analysis"

_geosciences, doi:10.3390/geosciences11080333_

Round 1

Reviewer 1 Report

The paper deals with the prediction of the shallow landslide susceptibility in landslide hazard and risk assessment. The Authors have discussed with great attention and wealth of details the items of the research. The tested area for the susceptibility modelling is Mesima basin in Calabria Region (South Italy).

In the study were considered eighteen predisposing factors that the more influential are: slope gradient, TWI, soil texture and lithology.  The results showed that LR method is valid and the prediction of the spatial occurrence of shallow landslides is satisfactory.

There are no particular prescriptions except the possibility of reducing the bibliography which appears too extensive and could be reduced.

The paper appears well constructed, well documented and the results encourage the application of the method in other areas with different lithology to verify its reliability.

Author Response

Dear Reviewer ,
for response to comments, Please see the attachment file.

Reviewer 2 Report

The paper is interesting /well written and the overall quality of this submission is high. Thus I suggest acceptance after some revision.

Comments - suggestions:

1. The discussion session is more or less a presentation of the results. I suggest to add some paragraphs about the limitation and the assumptions of the proposed method. For example :

- different scale of the factors (scale varies from 1:5000 to 1:250.000)

- global logistic regression assumes spatial stationarity of the relationship between landslide susceptibility and predisposing factors. See - among others - Feuillet et al 2014 ( Progress in Physical Geography) and Chalkias et al 2020 ( Bulletin of Engineering Geology and the Environment) for more details.

- mapping landslides as points and not as polygons.

- the classification of the final susceptibility map (Jenks classification) is critical for the evaluation. Did you try different classification schemes (eg. Standard Deviation or quantile...)

Moreover, in this section you have to add comments about the comparison of the results of this work with other previous analyses. (E.g. are the validation results in line with other similar works? / are the results for the specific study area expected....)

2. The creation of the "non-landslides" dataset in logistic regression analysis is very important for the evaluation of landslide susceptibility. Please explain how did you create this dataset (e.g. exploratory analysis). How many points it has?

[see - among others - Zhou et al 2016 and Zhang, et al 2019 .,(Nat. Hazards Earth Syst. Sci.) for details]

Moreover, I suggest adding in figure 3 the "Non-Landslides" dataset.

3. In the "Data" session I think that some information about the data is missing. For each causal factor, the readers need to know the format (raster / vector), the scale (or the cell size), the source, and the timeof the creation If there are differences in scale and time the authors have to explain why they accept these differences and how they potentially affect the analysis. Moreover, some details about the landslides inventory are missing ((typical dimensions - size/area - mechanism of creation).

4. I suggest adding in the Introduction section a paragraph about the advantages of the adopted method (logistic regression).

5. There is a lot of critique in the international literature about the use on non parametric tests (as Jacknife test) for the evaluation of the significance of predisposing factors. The authors have to add some comments about this, or alternative try some other approaches for this (see - among others - Polykretis et al 2018 (Natural Hazards). After these tests you might exclude some parameters from the analysis (as you mention in lines 102 - 104 of the manuscript)

6. LINE 431: Why did you choose jenks classification (it seems that STDev or quantile classification should be effective). Please consider to try (in this or in a future paper) these classification schemes and compare the resulted LS maps

7. In table 2 please replace "Nord" with the correct "North"

Author Response

(The authors gave the same response as above.)

Reviewer 3 Report

Dear Authors,

I read your paper, it is clear and well written, but I believe it does not add any significant advance to the actual knowledge about landslide susceptibility.

You simply applied a well known procedure (which is also known to be one of the weakest for LSM purpose) to a new case study.

I did not found any novelty in your paper, so I believe it should not be considered for publication.

Author Response

(The authors gave the same response as above.)

Round 2

Reviewer 3 Report

Dear Authors,

thanks for improving your paper.

I believe it can be considered for publication as a technical note, since it lacks of novelty to be published as a research paper